# Polyphenolic Nanomedicine Regulating Mitochondria REDOX for Innovative Cancer Treatment

**DOI:** 10.3390/pharmaceutics16080972

**Published:** 2024-07-23

**Authors:** Mingchuan Yang, Yufeng He, Qingqing Ni, Mengxue Zhou, Hongping Chen, Guangyun Li, Jizhong Yu, Ximing Wu, Xiangchun Zhang

**Affiliations:** 1Tea Research Institute, Chinese Academy of Agricultural Sciences, Hangzhou 310008, China; yangmingchuan@tricaas.com (M.Y.); qhyf2019@yeah.net (Y.H.); zhoumengxue@tricaas.com (M.Z.); thean27@tricaas.com (H.C.); fqcyzx@163.com (G.L.); zhangxc@tricaas.com (X.Z.); 2Department of Orthopedics, Shanghai General Hospital, Shanghai Jiao Tong University, Shanghai 200080, China; 13671733209@163.com; 3Hangzhou Academy of Agricultural Sciences, Hangzhou 310008, China; 4Anhui Province Green Food Collaborative Technology Service Center for Rural Revitalization, School of Biological and Food Engineering, Hefei Normal University, Hefei 230601, China

**Keywords:** polyphenol nanomedicine, ROS, mitochondria REDOX, drug delivery, anticancer

## Abstract

Cancer remains a highly lethal disease globally. The approach centered on REDOX-targeted mitochondrial therapy for cancer has displayed notable benefits. Plant polyphenols exhibit strong REDOX and anticancer properties, particularly by affecting mitochondrial function, yet their structural instability and low bioavailability hinder their utility. To overcome this challenge, researchers have utilized the inherent physical and chemical characteristics of polyphenols and their derivatives to develop innovative nanomedicines for targeting mitochondria. This review examines the construction strategies and anticancer properties of various types of polyphenol-based biological nanomedicine for regulating mitochondria in recent years, such as polyphenol self-assembly, metal–phenol network, polyphenol–protein, polyphenol–hydrogel, polyphenol–chitosan, and polyphenol–liposome. These polyphenolic nanomedicines incorporate enhanced features such as improved solubility, efficient photothermal conversion capability, regulation of mitochondrial homeostasis, and ion adsorption through diverse construction strategies. The focus is on how these polyphenol nanomedicines promote ROS production and their mechanism of targeting mitochondria to inhibit cancer. Furthermore, it delves into the benefits and applications of polyphenolic nanomedicine in cancer treatments, as well as the challenges for future research.

## 1. Introduction

Cancer remains a significant global health concern, with projections indicating 2,001,140 new cases and 611,720 deaths in the United States in 2024 [1]. While existing cancer treatments such as surgery, chemotherapy, radiotherapy, and immunotherapy have demonstrated some effectiveness, they often lack specificity and can lead to severe side effects [2,3,4]. Researchers are investigating targeted drug delivery systems using nanomaterials to enhance treatment precision [5,6,7,8,9]. The development of tailored nanomedicines, including metal, protein-based, and lipid-based nanoparticles, has shown promise in targeting tumor tissues, cells, and organelles. Despite advancements, challenges persist as many promising nanoparticles in preclinical studies have encountered obstacles in clinical development [5,10].

Various strategies are currently being developed to address the challenges of tumor treatment, with mitochondria emerging as a promising target for anticancer therapies. One of the major reasons is due to the frequent alteration of mitochondrial function in tumors, making it a reliable target [11]. The disruption of REDOX balance and the removal of restrictions on REDOX signaling in mitochondria result in malignant progression and the development of treatment resistance [12]. By enhancing antioxidant capacity to counteract the increase in reactive oxygen species (ROS), mitochondria can generate and tolerate high levels of ROS, which play diverse roles in signaling networks related to tumor proliferation, survival, and metastasis [13,14]. Moreover, multiple biochemical cascade reactions that trigger apoptosis and necrosis converge on mitochondria, with REDOX signals playing a crucial role in activating these cascades [15]. Since mitochondria are the main cellular source of ROS, REDOX-active compounds like polyphenols can target these organelles to regulate reactive oxygen species levels and their REDOX signals. Cancer cells with elevated levels of ROS possess a more significant antioxidant load and a delicate REDOX equilibrium compared to normal cells, rendering them more susceptible to drugs that induce oxidative stress [16,17].

Polyphenols, derived from plants, are well known for their antioxidant properties, but their pro-oxidative activity may actually play a more significant role in cancer treatment. Interestingly, polyphenols have the ability to selectively reduce the vitality of cancer cells in comparison to normal cells [18,19,20]. Studies indicate that polyphenols can rapidly stimulate the formation of ROS in cancer cells while sparing normal cells [19,20]. This imbalance between ROS production and the efficiency of antioxidant defense is a crucial factor. Polyphenols induce differential expression of antioxidant enzymes such as catalase and superoxide dismutase in cancer cells and normal cells, respectively [21]. In cancer cells, there is a decrease in antioxidant levels, increase in free radicals, and a disruption of mitochondrial membrane potential, whereas normal cells show elevated catalase levels and insensitivity to hydrogen peroxide [22]. Furthermore, certain studies have shown that the pro-oxidant effect of polyphenols on cancer cells is concentration-dependent [23,24,25]. Although high concentrations of polyphenols can lead to oxidative stress and harm, lower levels have been shown to protect against H_2_O_2_-induced damage in cancer cells. Therefore, maintaining optimal polyphenol levels in vivo is crucial for their anticancer properties. The ROS produced by polyphenols stem from their ortho-di/trihydroxy compound structure, which is prone to self-oxidation or catalytic oxidation by transition metal ions like Cu^2+^ and Fe^3+^ in biological systems. Numerous studies have demonstrated that polyphenols induce oxidative stress and apoptosis in various cancer cells by generating a significant amount of ROS [26,27,28]. Subsequent researches have indicated that these ROS primarily target mitochondria, resulting in mitochondrial damage and the activation of the mitochondrial apoptotic pathway. Examples of mitochondrial damage in cancer cells caused by polyphenol-induced ROS include decreased membrane potential, disruption of respiratory chain complexes, mitochondrial DNA damage, and the activation of caspase3 and caspase9 (Table 1).

Polyphenolic compounds have demonstrated potential in combating cancer, but their effectiveness is limited by poor pharmacokinetics, low stability, and restricted water solubility [29,30]. Therefore, the development of new drug delivery systems is crucial to enhance the stability and bioavailability of polyphenols, ensuring their concentration and biological activity in vivo. Recent advancements in nanotechnology have enabled the use of nanocarriers for encapsulating polyphenols [31,32,33,34,35]. These nanocarriers protect the encapsulated polyphenols from external physiological conditions, thereby preventing potential alterations in the composition and structure of bioactive compounds. This review will first discuss common methods for preparing high-performance nanomedicines using polyphenols, followed by an exploration of how these innovative polyphenolic nanomedicines disrupt the mitochondria of tumor cells. Lastly, it will outline potential future opportunities for the development and clinical application of polyphenolic nanomedicines.

**Table 1 pharmaceutics-16-00972-t001:** Summary of the mitochondrial apoptosis pathway activated via ROS generated by polyphenols.

Polyphenols	Mechanisms on REDOXResponsiveness	Mitochondrial Damage	Mitochondrial Apoptosis Pathway	Cancer Cell	Ref.
EGCG	ROS generation	Altered mitochondrial transmembrane potentials	Altered Bcl-2 family proteins, cytochrome C release, and activation of caspase 3 and caspase 9	Hepatocarcinoma SMMC7721 cells	[36]
Curcumin	superoxide anion O_2_^-^ production	Mitochondrial DNA damage, disruption of mitochondrial membrane potential	Cytochrome C release into the cytosol	HepG2 hepatocellular carcinoma cells	[37]
Dimethoxycurcumin	ROS generation	Reduced mitochondrial membrane potential	Decrease in cellular energy status (ATP/ADP)	Human breast carcinoma MCF7 cells	[38]
Resveratrol	ROS generation	Mitochondrial membrane potential loss	Decrease in glutathione levels, along with reduced mRNA expression and activity of superoxide dismutase	T cell leukemia Jurkat cells	[39]
trans-Resveratrol	Superoxide anions generation	—	Increasing in caspase 3-like activity	Human colorectal carcinoma HT-29 cells	[40]
Resveratrol derivatives	ROS generation	Mitochondrial depolarization	—	Colon cancer CT-26 cell	[41]
Quercetin	ROS generation	—	Elevated expression and activity of caspase 9	Human glioblastoma A172 cell	[42]
Oroxylin A	ROS generation	—	Elevated levels of SIRT3 in mitochondria, and the detachment of mitochondrial hexokinase II and the inhibition of glycolysis	Human breast carcinoma cell	[43]
Kaempferol	ROS generation	Altered mitochondrial membrane potential	Decreased expression of Bcl-2, elevated active caspase 3 and cleaved poly (ADP-ribose) polymerase expression	Human glioblastoma cells	[44]
Flavonoid LW-214	ROS generation	Mitochondrial membrane potential loss	Increased Bax/Bcl-2 ratio, caspase 9 activation, degradation of poly (ADP-ribose) polymerase (PARP), cytochrome C release and apoptosis-inducing factor transposition	Human breast cancer MCF-7 cells	[45]
Novel isoflavone derivative, NV-128	Superoxide and hydrogen peroxide generation	—	Decreasing in ATP, Cox-I, and Cox-IV levels	CD44+/MyD88+ ovarian cancer stem cells	[46]
Hesperidin	ROS generation	Mitochondrial membrane potential loss	Enhanced cytochrome C and apoptosis-inducing factor release from mitochondria, and caspase 3 activation.	HeLa cells	[47]
Naringenin	ROS generation	Mitochondrial depolarization	—	human epidermoid carcinoma A431 cells	[48]
Apigenin	ROS generation	Disruption of mitochondrial membrane potential	Glutathione depletion, cytosolic release of cytochrome C	Human prostate cancer 22Rv1 cells	[49]
luteolin	ROS generation	Mitochondrial membrane potential loss, and mitochondrial swelling	Release of cytochrome C	Hepatocellular carcinoma cells	[50]
Hesperetin	ROS generation	Reduced mitochondrial membrane potential	Increase in cytochrome C	Colon adenocarcinoma HT-29 cells	[51]
Chrysophanol	ROS generation	Reduced mitochondrial membrane potential	—	A549 human lung cancer cells	[52]
Agrimoniin	ROS generation	Disruption of mitochondrial membrane potential	—	Pancreatic cancer cells	[53]
3-deoxysappanchalcone	ROS generation	Mitochondrial membrane potential depolarization	—	Esophageal Squamous cell carcinoma ESCC cells	[54]
Calycosin	ROS generation	Reduced mitochondrial membrane potential	Decreased the expression of Bcl-2 and increased the expression of Bax, caspase 3, and poly (ADP ribose) polymerase	HepG2 hepatocellular carcinoma cells	[55]
Chlorogenic Acid	Superoxide (O_2_^•-^)	Reduced mitochondrial membrane potential, changes in mitochondrial morphology	—	MCF-7, MDA-MB-231, and HCC1419 breast cancer cells	[56]
Genistein	ROS generation	Decreased mitochondrial membrane potential, decrease mitochondrial activity	Up-regulated expression cytochrome C and Bax, decreased the expression of Bcl-2	Non-small lung cancer A549 and 95D cells	[57]
Gallic acid	ROS generation	Mitochondrial respiratory inhibition	Reduced ATP levels	Acute myeloid leukemia cells	[58]
Tannic acid	ROS generation	Reduced mitochondrial membrane potential	Reduced ATP levels, the activation of the death ligand TRAIL	Human embryonic carcinoma cells	[59]
Gossypol	ROS generation	Mitochondrial membrane potential loss,	Release of cytochrome C and apoptosis-inducing factor from mitochondria to the cytoplasm	human colorectal carcinoma cells	[60]

## 2. Polyphenolic Nanomedicine

The utilization of polyphenols in nanomaterial synthesis has demonstrated effectiveness in enhancing stability and retention time, while also maintaining their REDOX capabilities. Polyphenols with catechol and gallic groups demonstrated involvement in hydrogen bonding, hydrophobic interactions, and π-π interactions [61]. Several studies have emphasized these attributes of polyphenols as a strong foundation for creating versatile nanomaterials [31]. This article offers a detailed exploration of the potential anticancer properties of polyphenolic nanomaterials specifically engineered to target mitochondria (Figure 1).

The mechanism by which polyphenols induce cancer cell apoptosis through the production of ROS targeting mitochondria is well understood. However, challenges such as limited water solubility and bioavailability hinder the achievement of optimal concentrations in the blood and at the tumor site for treatment [62]. Polyphenol nanomedicines address these limitations by enhancing water solubility, bioavailability, and pharmacokinetics in vivo [63]. Importantly, polyphenol nanomedicines can accumulate in large amounts at the tumor site [64]. These nanomedicines sustain the REDOX activity of polyphenols upon reaching cancer cells, leading to the generation of significant ROS and mitochondrial impairment [65,66]. Furthermore, the release of Fe^3+^ or Cu^+^ from polyphenol nanomedicines can enhance the Fenton reaction in cells, resulting in increased ROS levels [64,67]. Polyphenol nanomedicines with photothermal properties have synergistic therapeutic effects, such as inducing ROS generation in tumors under near-infrared (NIR) laser irradiation [68,69]. In conclusion, the advantages of polyphenol nanomedicines over pure polyphenols include promoting polyphenol accumulation at tumor sites, ultimately inducing cancer cell apoptosis through ROS targeting mitochondria.

### 2.1. Polyphenol Self-Assembly Nanomedicine

The presence of catechol and/or galloyl groups in polyphenols allows for a variety of covalent and non-covalent interactions to contribute to the formation of polyphenol-based materials [31]. For example, dopamine, containing both catechol and amine groups, readily undergoes oxidation and self-polymerization to create polydopamine (PDA) [70]. Similarly, (-)-Epigallocatechin-3-gallate (EGCG) has been observed to form self-assembled nanoparticles, exhibiting enhanced stability and REDOX activity characteristics compared to native EGCG in both in vitro and in vivo scenarios [71]. Peng et al. [72] also confirmed the existence of self-assembled nanoparticles of polyphenols with pectin, maintaining a stable system in chrysanthemum tea infusion. These self-assembled nanomaterials of polyphenols have been utilized in anticancer therapies due to their REDOX activity, high drug loading capacity for hydrophobic and hydrophilic agents, and photothermal properties [73]. Nieto et al. [74] found that PDA nanoparticles can induce apoptosis in breast cancer cells by stimulating ROS production while maintaining non-toxicity towards normal cells. Moreover, PDA has exhibited efficient conversion of near-infrared light into heat, showcasing its potential as a photothermal agent for eliminating cancer cells both in vitro and in vivo [75,76] (Figure 2A,B). The aromatic rings present in PDA allow for the loading of chemical drugs or dyes on its surface through π-π stacking and/or hydrophobic–hydrophobic interactions. In a study by Meng et al. [77], a mitochondria-targeting nanoparticle was designed to deliver PDA as a photothermal agent and alpha-tocopherol succinate as a chemotherapeutic drug to tumor cell mitochondria, resulting in cellular apoptosis and a synergistic inhibition of tumor cell proliferation. Furthermore, Chen et al. [78] developed a novel aggregation-induced emission (AIE) gen (MeO-TPE-indo, MTi) and incorporated it onto the surface of PDA to create the nanocomposite (PDA-MeO-TPE-indo, PMTi), which could effectively generate ROS for photodynamic therapy and target mitochondria. Moreover, the self-assembly of polyphenols with other substances holds promise in anticancer applications. For example, a nanosystem containing ursolic acid and EGCG demonstrated favorable outcomes in the treatment of hepatocellular carcinoma without causing adverse effects on normal tissues [79] (Figure 2C).

### 2.2. Metal-Phenolic Network Nanomedicine

Polyphenols are widely present in nature, with over 8000 known organic species that can coordinate with metal ions to form metal–phenolic network nanoparticles (MPNs) [80,81,82]. The rapid chelation between phenolic ligands and metal ions allows for the quick formation of MPNs coatings within minutes [82,83]. By adjusting the pH during the assembly of MPNs, targeted tumor localization can be achieved [84]. MPNs can promptly attach to individual cancer cells [85], leading to long-term antitumor effects by regulating ROS and REDOX processes [86]. These MPNs hold promise in cancer treatment due to the catalytic activity of metal ions and the REDOX activity of polyphenols [87]. Li et al. [88] introduced a combination chemotherapy platform that depletes ATP and enhances ROS by integrating therapeutic samarium (Sm^III^) ions and (-)-epicatechin (EC) through a metal-phenolic network (Sm^III^–EC) (Figure 3A). The resulting Sm^III^–EC nanoparticles not only reduced tumor volume but also did not affect the body weight of mice or normal organs when compared to the clinical anticancer drug 5-fluorouracil for colon cancer treatment. Zhao et al. [67] developed MPNs using copper ions and gallic acid (Cu-GA) to induce apoptosis and cuproptosis for synergistic chemo/chemodynamic therapy. The release of gallic acid significantly reduces intracellular GSH content, making tumor cells unable to scavenge ROS and more susceptible to cuproptosis. Furthermore, the Fenton-like reaction of released Cu^+^ increases ROS levels within tumor cells, disrupting REDOX homeostasis and leading to apoptosis-related chemodynamic therapy. Liao et al. [89] created lanthanide metal ions (Sm^III^)–EGCG networks for targeted therapy against metastatic melanoma, and the synergistic effects of Sm^III^–EGCG networks demonstrate remarkable tumor inhibition properties compared to the clinical anticancer drug, 5-fluorouracil.

MPNs also function as a versatile nanoplatform for enhancing interfacial adhesion and drug loading in cancer therapy. Meng et al. [90] developed a multifunctional nanocomposite (PID@Fe-TA) that combines chemodynamic therapy, photothermal therapy, and chemotherapy to effectively combat breast cancer, offering a promising approach for cancer treatment. Dai et al. [91] created biocompatible MPNs to encapsulate DOX, where DOX coordinates with Fe^3+^, leading to the formation of DOX@Pt prodrug Fe nanoparticles (DPPF NPs), which act as a ROS enhanced combination chemotherapy platform (Figure 3B). Additionally, EGCG-based MPNs showed potential for incorporating DOX to achieve enhanced synergistic chemotherapy [92,93], due to the EGCG-mediated downregulation of CBR1 protein, which inhibits the generation of doxorubicinol (DOXOL), a reduction product of DOX linked to drug resistance and cardiotoxicity.

**Figure 3 pharmaceutics-16-00972-f003:**
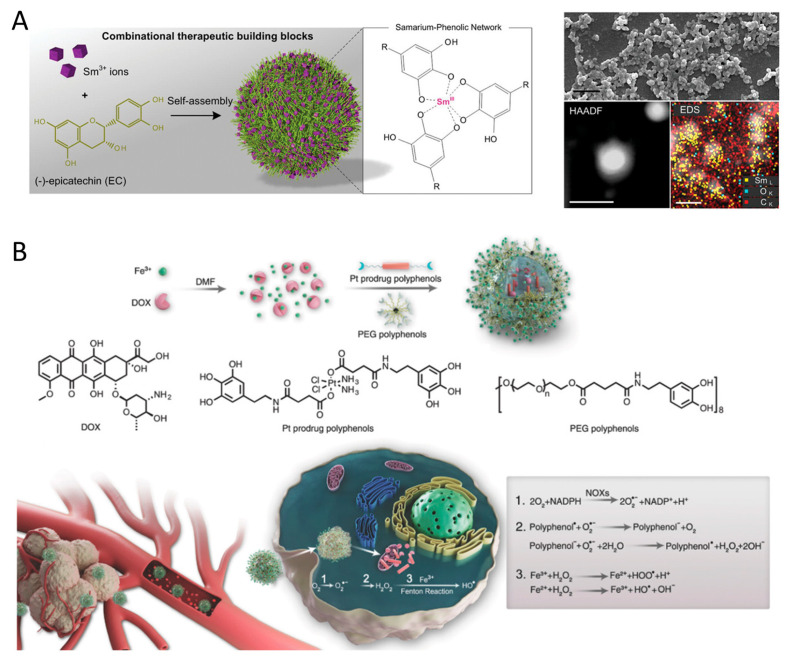
(**A**) Self-assembly of the Sm^III^-EC nanoparticles from two therapeutic building blocks, rare earth Sm^3+^ ions and phenolic EC molecules. Reproduced with permission [88]. Copyright 2019, WILEY. (**B**) Formulation of DOX@Pt prodrug Fe nanoparticles (DPPF NPs) and the ROS enhanced chemotherapy mechanism in the tumor cells. NOXs were activated by DOX and platinum drugs and catalyzed the O_2_ to O_2_^•−^ SOD-like activity of polyphenols catalyzes H_2_O_2_ generation from O_2_^•−^. Ferric ions further turn H_2_O_2_ into highly toxic HO^•^ by Fenton reaction. Reproduced with permission [91]. Copyright 2018, WILEY.

### 2.3. Polyphenol–Protein Nanomedicine

Proteins have been extensively studied in cancer therapy, due to their biocompatibility, biodegradability, and non-immunogenicity [94,95]. Although the stability and intracellular delivery efficiency of proteins are limited, custom-designed protein-based nanosystems have shown promise in addressing these challenges. Polyphenols have been shown to interact with proteins through covalent and non-covalent bonds, facilitating cross-linking between proteins and improving delivery efficiency [96] (Figure 4A). For example, Qiao et al. [97] developed an ROS amplifying nanodevice by directly complexing melittin and condensed EGCG, and then prepared pHA-NC by covering it with phenylboronic acid derivatized hyaluronic acid (HA) (Figure 4B). Upon receptor-mediated endocytosis, pHA-NC increased intracellular oxidative stress, resulting in enhanced anticancer efficacy in cancer cells. To enhance the dissolution rate and aqueous solubility of curcumin, Jithan et al. [63] developed curcumin nanoparticle using bovine serum albumin. Curcumin nanoparticle showed superior anticancer effects in MDA-MB-231 cells compared to free curcumin, along with increased bioavailability and enhanced pharmacokinetic properties in rats. To investigate the potential antitumor effects of EGCG nanoparticles, Yang et al. [98] used thermally modified β-lactoglobulin, 3-mercapto-1-hexanol, and EGCG to create stable co-assembled nanocomplexes (MEβ-NPs). These MEβ-NPs showed no toxicity in mice but exhibited a strong inhibitory effect on the growth of implanted human melanoma A375 cell tumors, proving to be twice as effective as free EGCG. Additionally, a more advanced delivery system, micellar nanocomplexes (MNCs), was developed by combining oligomeric EGCG with the anticancer protein Hesetin to create a core, and then combining poly (ethylene glycol)-EGCG to form the shell [99]. The MNCs demonstrated enhanced tumor selectivity, improved growth inhibition, and an extended blood half-life. This EGCG-protein nanomedicine holds promise for phototherapy in cancer treatment. Wang et al. [68] developed nanoparticles S-aPDL1/ICG@NP by combining anti PD-L1 and photosensitizer ICG with EGCG dimer and MMP-2-susceptible blocking polymer PEG-PLGLAG-dEGCG through non-specific hydrophobic and electrostatic interactions. The S-aPDL1/ICG@NP facilitated the generation of intratumoral ROS under NIR laser irradiation. Furthermore, S-aPDL1/ICG@NP enhanced the intratumoral infiltration of cytotoxic T cells, effectively suppressing growth and metastasis.

### 2.4. Polyphenol-Hydrogel Nanomedicine

Hydrogel is a 3D polymer network formed by cross-linking hydrophilic macromolecular chains in an aqueous environment [100]. It provides a promising approach for delivering therapeutic payloads due to its biocompatibility and structural similarity to the natural extracellular matrix [101]. Polyphenol compounds containing groups such as catechol and pyrogallol readily interact with other substances to form 3D hydrogel networks with exceptional properties like adhesion, toughness, photothermal effects, antibacterial, and anticancer properties [102]. Teong et al. [103] synthesized several types of curcumin-hydrogel nanoparticles using curcumin, biopolymeric chitosan, gelatin, and hyaluronan nanoparticles in an electrostatic field system. These curcumin–hydrogel nanoparticles increased apoptosis in A549 cells by stimulating ROS production and reducing mitochondrial membrane potential levels, demonstrating higher anticancer efficacy than curcumin alone. Furthermore, Madeo et al. [104] prepared graphene oxide-loaded curcumin nanosheets encapsulated in hydrogels, cross-linked with Ca^2+^, to form hybrid hydrogels used as patches for the local treatment of areas affected by squamous cell carcinoma (Figure 5A). The curcumin-loaded systems exhibited potent cytotoxic effects in SCC cancer cells, with sustained release of curcumin (~50% after 96 h). Jeong et al. [105] developed Au ion-crosslinked hydrogels (ICGs) containing indocyanine green and EGCG for photothermal/photodynamic therapy in breast cancer treatment. ICGs not only produced ROS but also induced hyperthermia for photothermal therapy upon exposed to NIR laser, effectively inhibiting the growth of primary breast cancer. He et al. [106] created a palladium single atom nanoenzyme supported by polyphenol modified carbon quantum dots (DA-CQD@Pd SAN), and used it to fabricate a bioadhesive hydrogel for local tumor immunotherapy (Figure 5B). This hydrogel not only locally delivers immune adjuvants but also enhances the antitumor effect by scavenging a significant amount of ROS generated by H_2_O_2_ within the tumor.

Polyphenol hydrogels can be enhanced by combining them with chitosan or liposome to improve stability and drug release properties. George et al. [107] used l-histidine coupled chitosan, plant synthesized zinc oxide nanoparticles and dialdehyde cellulose to prepare functional nano hybrid hydrogel (HIS-CHGZ) for delivering naringenin, quercetin, and curcumin. The drug release from HIS-CHGZ-polyphenol followed a non-Fickian diffusion-based mechanism along with polymer erosion, resulting in a significant 15-to-30-fold increase in cytotoxicity of HIS-CHGZ-polyphenol in A431 cells compared to free polyphenols. Li et al. [108] developed a composite hydrogel (Cur-lip/DOX/CSSH) containing thiolated chitosan (CSSH), DOX, and hydrophobic liposome-encapsulated curcumin (Cur-Lip) (Figure 5C). The Cur-Lip/DOX gels exhibit long-term drug release capability and potent anticancer activity. Furthermore, polyphenol hydrogel can be integrated with 3D printing. Zhu et al. [109] synthesized a multifunctional PC@ZIF-8@PDA/SerMA hydrogel using methacrylic anhydride (MA)-modified sericin (SerMA) solution mixed with procyanidin (PC)-loaded PDA-modified ZIF-8 (PC@ZIF-8@PDA) nanoparticles under blue light. PC@ZIF-8@PDA/SerMA hydrogels with personalized shapes fabricated through 3D printing demonstrated the ability to eliminate chondrosarcoma cells through their photothermal activity in conjunction with NIR radiation.

**Figure 5 pharmaceutics-16-00972-f005:**
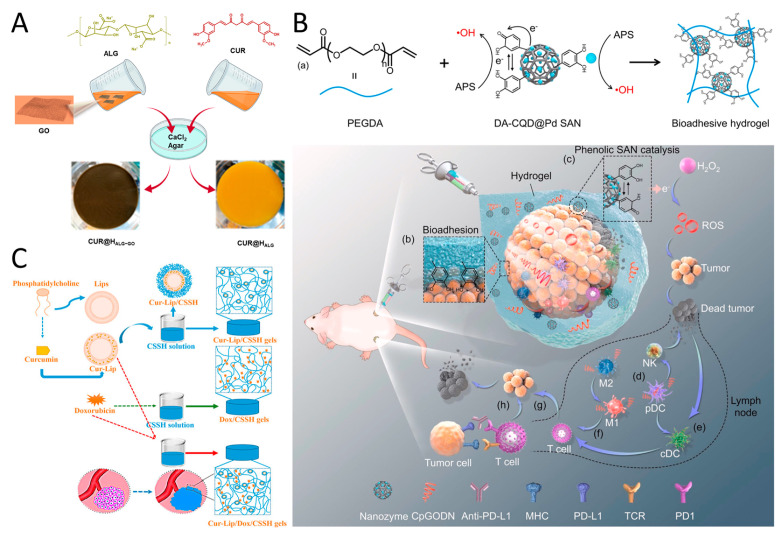
(**A**) Curcumin (CUR) was loaded onto graphene oxide (GO) nanosheets, mixed into alginate gel crosslinked by calcium ions, resulting in a hydrogel patch for treating squamous cell carcinoma (SCC). Reproduced with permission [104]. Copyright 2021, Elsevier. (**B**) DA-CQD@Pd@CpGODN hydrogel induces tumor death by generating ROS and treats distant untreated tumors through catalytic immunotherapy. (a) In situ hydrogel gelation by the SAN catalyzing. (b) Hydrogel was injected and adhered at tumor sites. (c) The catalytic process of the DA-CQD@Pd SAN in the TME. The hydrogel induced activation of (d) pDC, (e) cDC, and (f) T cells in TME. (g) T cells triggered the adaptive immune response. (h) Anti-PD-L1 enhanced the immune response by blocking the binding of PD-L1 to PD1 on T cells. Reproduced with permission [106]. Copyright 2022, Elsevier. (**C**) A smart, temperature-responsive CSSH hydrogel was designed to facilitate the in situ-coating of solid tumors, repair the leaky vasculature and impaired lymphatic drainage, and fill defects after tumor excision. Reproduced with permission [108]. Copyright 2022, Frontiers Media.

### 2.5. Polyphenol–Chitosan Nanomedicine

Chitosan, a naturally derived polymer produced through alkaline deacetylation of chitin, is recognized for its non-toxic, biocompatible, and biodegradable properties [110]. Due to its abundant functional amino and hydroxyl groups, chitosan readily binds to other active molecules [111,112]. Polyphenols can form stable conjugates with chitosan through mechanisms such as hydrogen bonding, hydrophobic interactions, and van der Waals forces [113]. Various types of polyphenol–chitosan conjugates have demonstrated enhanced solubility, sustained release, and improved bioavailability [114,115] (Figure 6A). For example, Mariadoss et al. [116] developed chitosan nanoparticles loaded with phloretin (PhCsPs) to investigate their potential in oral cancer treatment. These nanoparticles enhanced the mitochondrial-mediated apoptotic mechanism by inducing ROS generation in KB cells. Similarly, Tsai et al. [117] created nanoparticles (CENP) containing curcumin by crosslinking chitosan and tripolyphosphate for photodynamic therapy (Figure 6B). The CENP nanoparticles exhibited enhanced photodynamic therapy in cancer cells, producing significantly more ^1^O_2_, resulting in an approximately fourfold decrease in the IC50 value. Khan et al. [118] synthesized an oral formulation of chitosan-based nanoparticles loaded with EGCG for the sustainable and controlled release of polyphenol to combat prostate cancer. The EGCG–chitosan significantly suppressed tumor growth and prostate-specific antigen levels compared with free EGCG in 22Rν1 tumor xenografts implanted in athymic nude mice. The enhanced anticancer efficacy of EGCG–chitosan is attributed to its prolonged release, as encapsulated EGCG exhibited slow discharge from the nanoparticles.

Polyphenol–chitosan has demonstrated potential not only as an anticancer agent, but also as a drug delivery system for other active ingredients. Hu et al. [119] synthesized polymer nanoparticles (GA-g-CS-CPP) made from gallic acid grafted chitosan and caseinophosphopeptides to enhance the delivery efficiency and resistance to degradation of EGCG nanoparticles. Moreover, polyphenol–chitosan has been utilized for delivering chemotherapy drugs. Wang et al. [120] developed an amphiphilic carboxymethyl chitosan–quercetin (CQ) to improve oral bioavailability of paclitaxel (PTX). The PTX-loaded CQ exhibited sustained release in simulated gastrointestinal fluid and enhanced paclitaxel permeability in intestinal absorption experiments. Compared to Taxol (^®^) and Taxol (^®^) combined with verapamil, PTX-loaded CQ showed potent antitumor efficacy in tumor xenograft models. Similarly, a novel pH-responsive nanomicelle (QT-CA-CS) was designed using chitosan, quercetin, and citraconic anhydride to encapsulate the anticancer drug DOX [121]. The QT-CA-CS-DOX nanomicelles improved the cellular uptake of DOX in drug-resistant MCF-7/ADR cells by 8.62 times compared to free DOX. Notably, these nanomicelles could evade lysosomes, leading to the rapid release of DOX and quercetin in the cytoplasm, resulting in a stronger inhibitory effect on tumor cells. Recently, Mu et al. [122] introduced a delivery system, Cabazitaxel (Cab)@MPN/CS, comprising metal–polyphenol and chitosan, for melanoma therapy. (Cab)@MPN/CS exhibited prolonged retention in tumor tissue and effective tumor suppression.

### 2.6. Polyphenol–Liposome Nanomedicine

Liposomes, nanoscale artificial vesicles composed of lipid bilayers encapsulating an aqueous core [123], mimic natural cell membranes and possess biocompatible and biodegradable properties, making them ideal carriers for drug delivery. One notable characteristic of liposomes is their ability to encapsulate hydrophobic drugs in the lipid layer and hydrophilic drugs in the internal water core, protecting them from degradation and metabolism in the body’s circulation [124,125]. Given this feature, various plant polyphenols have been loaded into liposomes to create polyphenol–liposome complexes with enhanced solubility and chemical stability [126]. Caddeo et al. [65] developed eudragit-coated liposomes to safely transport resveratrol and artemisinin through the gastrointestinal tract, targeting the intestine. These vesicles exhibited pro-oxidant effects in intestinal adenocarcinoma cells, resulting in significant cell death by inducing mitochondrial ROS production. To achieve extended blood circulation and improve permeability and retention, Kang and Ko [66] constructed mitochondria-targeting liposomes loaded with resveratrol by surface modification with TPP-PEG or DQA-PEG. These resveratrol–liposomes exhibited improved cellular uptake and specific accumulation in the mitochondria. Furthermore, they induced cancer cell cytotoxicity through ROS generation and dissipation of mitochondrial membrane potential. Additionally, in a cisplatin-resistant human ovarian tumor xenograft model, quercetin–liposome and honokiol–liposome exhibited significant suppression of tumor growth compared to free polyphenols [127,128].

Polyphenol liposomes can be combined with chitosan or hydrogel to enhance the bioavailability of polyphenols. Ezzat et al. [129] detailed the creation of catechin-loaded chitosan-tethered liposomes (CHS) through the ethanol injection technique to enhance the oral bioavailability of catechin. Pharmacokinetic studies in rats demonstrated that catechin liposomes increased the levels and duration of catechins in plasma compared to free catechins. Li et al. [130] developed curcumin liposome (CurLip), and coated it with CSSH to create liposomal hydrogels (CSSH/CurLip gel), which improved the solubility and bioavailability of curcumin. The CSSH/Cur-Lip gel inhibited breast cancer recurrence post-tumor resection and promoted tissue repair of the defect. Wu et al. [64] recently introduced an inhalable gallic acid–Fe MPN hybrid liposome (LDG) to enhance the intracellular Fenton reaction (Figure 7). LDG demonstrated excellent nebulization capability, significantly increasing lung accumulation. Furthermore, LDG initiated a strong Fenton response, resulting in intracellular ROS generation, and displaying significant antitumor efficacy in an orthotopic lung tumor model.

### 2.7. Other Polyphenolic Nanomedicine

Other nano carriers for polyphenol delivery, such as nanoemulsions and nanogels, have also been developed to cater different anticancer strategies. Nanoemulsion, a distinctive structure consisting of an oil phase, water phase, and surfactant or emulsifier, is regarded as a delivery system for to enhance the effectiveness of hydrophobic antitumor drugs [131]. It can improve drug dosage efficacy, stability, and demonstrate slow release and targeting effects. For example, quercetin-nanoemulsion has shown improved encapsulation efficiency, and enhanced antitumor effects compared to free quercetin [132]. Nanogel has also emerged as a promising nano carrier for targeted delivery of therapeutic agents in cancer treatment [133]. With its porous structure and large surface-to-volume ratio, nanogel can enhance drug permeability and retention at the tumor site, thereby improving therapeutic outcomes. A curcumin-loaded nanogel has achieved sustained drug release, resulting in enhanced suppression of cancer cells and tumor growth compared to free curcumin [134].

## 3. Conclusions and Outlooks

This review summarizes recent advancements in utilizing polyphenol molecules as the fundamental components for developing a range of biocompatible nanomaterials. The study indicates that these biocompatible nanomaterials, formed through self-assembly, MPN formation, covalent bonding, and hydrophobic/hydrogen bonding interactions, exhibit anticancer properties with targeting mitochondria. Furthermore, these polyphenol nanomedicines have also acted as nanocarriers for delivering various therapeutic drugs to tumors, including different metal ions, chemotherapy drugs, and photothermal agents.

Polyphenol nanomedicines are typically prepared by combining different materials to enhance their functionality. However, the intricate nature of nanomedicines poses challenges in terms of body metabolism. Future research on polyphenol nanomedicines should concentrate on developing simple and versatile preparation techniques to achieve rational design and customizable synthesis. The self-assembly of polyphenolic metals and chemical grafting techniques may provide promising avenues for further exploration. Furthermore, combination therapy appears to better reflect the therapeutic characteristics of polyphenol nanomedicines compared to single therapies. Understanding how polyphenols, chemotherapy, phototherapy, and other drugs interact within in cancer cells can help in leveraging the characteristics of polyphenolic nanomedicines. Polyphenol nanoparticles, with their photothermal properties and the ability to overcome drug resistance, hold significant promise when combined with treatments like radiotherapy and immunotherapy. It is essential to integrate nanotechnology with other scientific disciplines to broaden treatment options such as transdermal agents, inhalable drugs, and photothermal therapy, to meet the specific needs of patients. Future research on polyphenolic nanomedicines should prioritize their REDOX activity for precise targeting of tumor cell mitochondria.

## Figures and Tables

**Figure 1 pharmaceutics-16-00972-f001:**
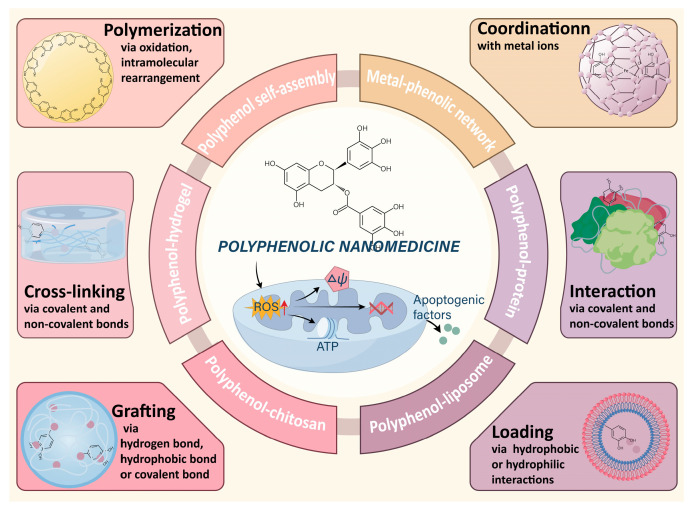
Schematic diagram of the construction of polyphenolic nanomedicine through various typical chemical materials.

**Figure 2 pharmaceutics-16-00972-f002:**
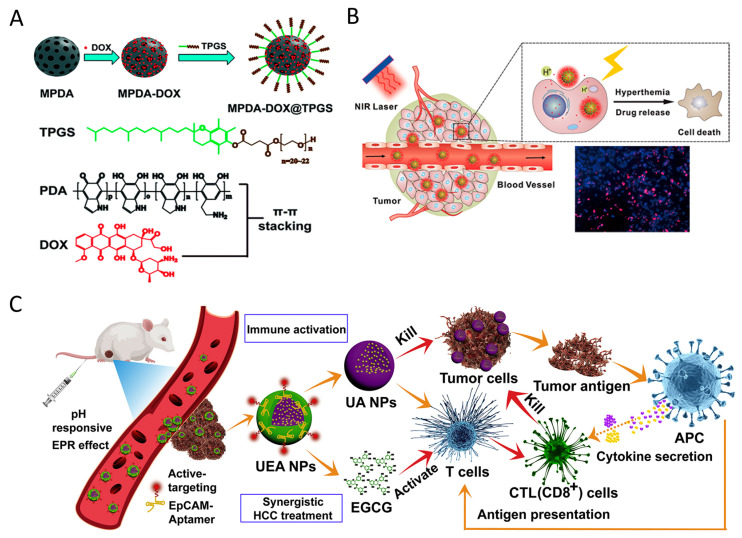
(**A**) Doxorubicin hydrochloride (DOX) and d-α-tocopheryl polyethylene glycol 1000 succinate (TPGS) were sequentially loaded in the pore space and on the external particle surface of mesoporous polydopamine nanoparticles (MPDA) via π-π stacking and hydrophobic interactions, resulting in a MPDA–DOX@TPGS complex. Reproduced with permission [75]. Copyright 2017, Royal Society of Chemistry. (**B**) Polydopamine nanoparticles release anticancer drugs upon stimulation by NIR light, pH, and ROS, leading to cancer cell apoptosis. Reproduced with permission [76]. Copyright 2016, Elsevier. (**C**) Synergistic hepatocellular carcinoma treatment of the “carrier-free” Apt-modified nanodrug based on the ursolic acid and EGCG by activating the innate and acquired immunity. Reproduced with permission [79]. Copyright 2021, Elsevier.

**Figure 4 pharmaceutics-16-00972-f004:**
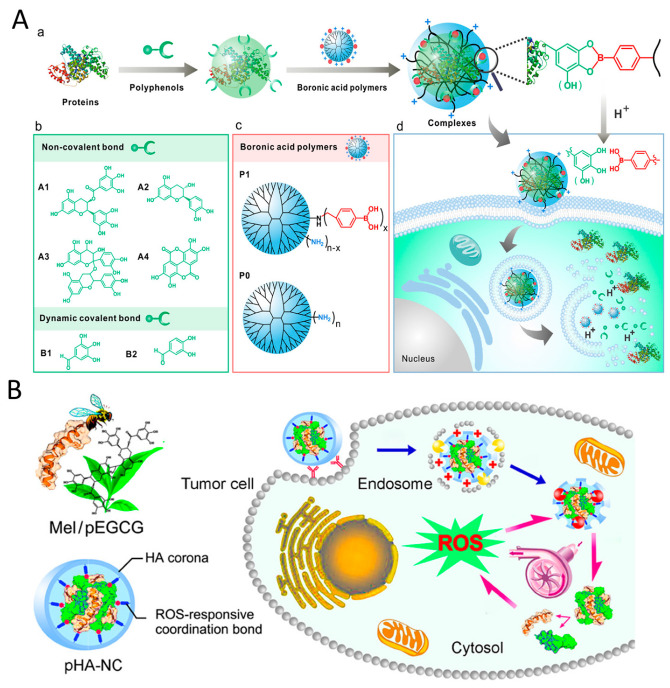
(**A**) Polyphenol-mediated cytosolic protein delivery by boronic acid-decorated polymers. (**a**) Polyphenol facilitates the formation of protein transduction complexes. (**b**) Structures of the investigated polyphenols. (**c**) Structures of the boronic acid-decorated polymer P1 and the control polymer PO. (**d**) Intracellular release of proteins from the nanoparticles triggered by lysosomal acidity. Reproduced with permission. [96] Copyright 2019, ACS Publications. (**B**) Schematic illustration of a multipronged nanocomplex (pHA-NC) comprising mel and pEGCG surrounded by HA through the coordination bond for ROS self-sufficient oxidation therapy of cancers. Reproduced with permission. [97] Copyright 2018, ACS Publications.

**Figure 6 pharmaceutics-16-00972-f006:**
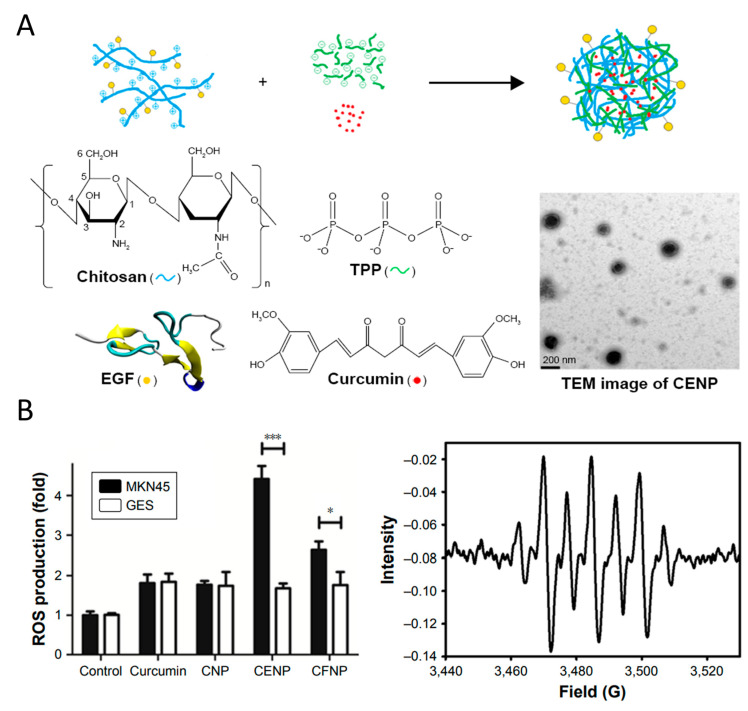
(**A**) A schematic diagram for the nanoparticle assembly of curcumin-encapsulated and EGF-conjugated chitosan/TPP nanoparticles (CENP) from EGF-conjugated chitosan, TPP, and curcumin. (**B**) ROS generated in photodynamic therapy treated with various nanoparticles include the production of free radicals and _1_O^2^, * *p* < 0.05, *** *p* < 0.001. Reproduced with permission [117]. Copyright 2017, Dove Medical Press Ltd.

**Figure 7 pharmaceutics-16-00972-f007:**
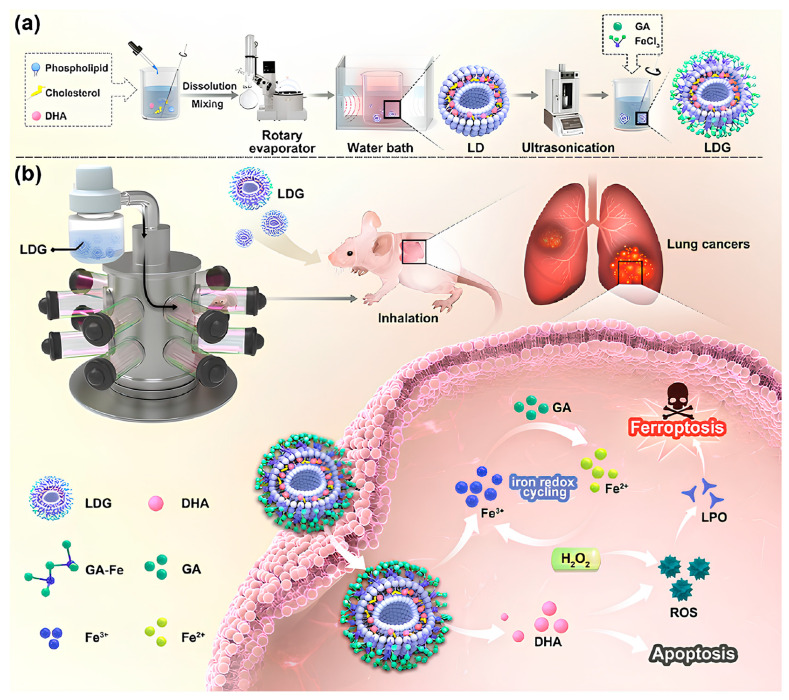
MPN hybrid liposomes for lung cancer ferroptosis-apoptosis synergetic therapy. (**a**) Gallic acid–Fe MPN hybrid liposome (LDG) liposomes were prepared by thin film ultrasonic dispersion. (**b**) Nebulization delivered antitumor strategy of LDG based on iron REDOX cycling. Reproduced with permission [64]. Copyright 2024, Tsinghua University Press.

## Data Availability

Data derived from public domain resources. The data presented in this study are available in reference numbers [64,75,76,79,88,91,96,97,104,106,108,117].

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
