# Peer review of "Polyphenolic Nanomedicine Regulating Mitochondria REDOX for Innovative Cancer Treatment"

_pharmaceutics, 2024, doi:10.3390/pharmaceutics16080972_

Round 1

Reviewer 1 Report

Comments and Suggestions for Authors

The overall paper is very well written, though it could better describe exactly how these nanomedicines kill cancer cells. However, the figures appear to be taken from other manuscripts and are too complicated and the text too small to read. Figures need to be simplified and redone specifically for this manuscript.   

Author Response

Comments 1: The overall paper is very well written, though it could better describe exactly how these nanomedicines kill cancer cells. However, the figures appear to be taken from other manuscripts and are too complicated and the text too small to read. Figures need to be simplified and redone specifically for this manuscript.

Response 1: We thank the reviewer for this suggestion. We have described how these nanomedicines kill cancer cells . The figures have been simplified and redone specifically.

Reviewer 2 Report

Comments and Suggestions for Authors

Yang et. al. presented a review on ‘Polyphenolic nanomedicine regulating mitochondria RE[1]DOX for innovative cancer treatment’. For acceptance, this manuscript must undergo substantial revision.

1)     The abstract requires a rewrite with a clear focus.

2)     The clarity of the figures needs improvement. Reading it is a challenge.

3)     Additional details are needed to explain the metal phenolic network.

4)     What is Outlook? It is poor.

5)     Include a dedicated section outlining the mechanism in this manuscript.

Comments on the Quality of English Language

Need to revise.

Author Response

Comments 1: Yang et.al. presented a review on ‘Polyphenolic nanomedicine regulating mitochondria REDOX for innovative cancer treatment’. For acceptance, this manuscript must undergo substantial revision.

  1. The abstract requires a rewrite with a clear focus.

    Response 1: We thank the reviewer for this suggestion. The abstract has been rewritten for a clear focus. 

    Comments 2: The clarity of the figures needs improvement. Reading it is a challenge.

    Response 2: We thank the reviewer for this suggestion. These figures have been improved.

    Comments 3Additional details are needed to explain the metal phenolic network.

    Response 3: We thank the reviewer for this suggestion. Polyphenols are abundant in nature with over 8000 known organic species that can coordinate with metal ions in viro/vivo to form metal-phenolic network nanoparticles (MPNs) [78-80]. The rapid chelation between phenolic ligands and metal ionsenables the formation of MPNs coatings within minutes [80, 81]. By altering the assembly pH of MPNs can realizing tumor target localization [82]. MPNs can immediately adhere to the surface of individual cancer cells [83], resulting the long-term antitumor behaviors involving regulating ROS and redox [84]. This section is added to Metal-phenolic network nanomedicine in line 87-94, page 9.

    Comments 4: What is Outlook? It is poor.

    Response 4: We thank the reviewer for this suggestion. This section has been revised. “Polyphenol nanomedicines preparation typically involve combining different materials to enhance functionality. However, the complexity of nanomedicines presents challenges in body metabolism. Future studies on polyphenol nanomedicines should focus on creating uncomplicated and versatile preparation techniques for attaining ra-tional design and customizable synthesis. The self-assembly of polyphenolic metals and chemical grafting techniques may offer promising avenues for exploration. In ad-dition, compared to single polyphenol nanomedicine, combination therapy seems to better reflect their therapeutic characteristic. Understanding how polyphenols, chemo-therapy, phototherapy, and other drugs work in cancer cells can aid in comprehend-ing how to leverage the characteristics of polyphenolic nanomedicines. Polyphenol nanoparticles, with their photothermal properties and the ability to overcome drug re-sistance, could hold significant promise when combined with treatments like radio-therapy and immunotherapy. Nanotechnology should be integrated with other scien-tific disciplines to expand treatment options such as transdermal agents, inhalable drugs, and photothermal therapy, to meet the specific treatment needs of patients. Fu-ture research on polyphenolic nanomedicines should prioritize their redox activity for precise targeting of tumor cell mitochondria.”

    This section is added to Conclusion and outlooks in line 374-390, page 21.

    Comments 5: Include a dedicated section outlining the mechanism in this manuscript.

    Response 5: We thank the reviewer for this suggestion. The mechanism by which polyphenols induce cancer cell apoptosis through the production of ROS targeting mitochondria is well understood. However, challenges such as limited water solubility and bioavailability, hinder the achievement of optimal concentrations in the blood and at the tumor site for treatment[61]. Polyphenol nano-medicines addresses these limitations of polyphenols, enhancing water solubility, bio-availability, and pharmacokinetics in vivo [62]. More importantly, polyphenol nano-medicines can accumulate in large amounts at the tumor site[63]. Polyphenol nano-medicines sustain the redox activity of polyphenols upon reaching cancer cells, lead-ing to the generation of significant ROS and mitochondrial impairment[64,65]. Addi-tionally, the release of Fe3+ or Cu+ from polyphenol nanomedicines can enhance the Fenton reaction in cells, resulting in increased ROS levels[63,66]. Polyphenol nano-medicines with photothermal properties have the capability to induce ROS generation in tumors under near-infrared laser irradiation[67]. In conclusion, the advantages of polyphenol nanomedicines over pure polyphenols are promoting polyphenol accumu-lation at tumor sites, ultimately inducing cancer cell apoptosis through ROS targeting mitochondria.

    This section is added to 2. Polyphenolic nanomedicine in line 14-28, page 7.

Reviewer 3 Report

Comments and Suggestions for Authors

No

Author Response

Comments 1: No.

Response 1: We thank the reviewer for your recognition of the quality of the paper. 

Reviewer 4 Report

Comments and Suggestions for Authors

In this review the authors discuss the encapsulation strategies of polyphenols in nanocarriers with the aim of improve the stability and bioavailability of polyphenols, thus enhancing their antitumor effect by disrupting the mitochondria of tumor cells.

Although there are already other reviews on the topic, this one is focused on the regulation of RE-DOX mitochondria. It is interesting and could be published in this journal but should be enhanced by major revisions.

 Specific comments:

 -          In the introduction section authors reported “Cancer cells with higher intrinsic levels of reactive oxygen species have a more sensitive antioxidant capacity than normal cells, making them less resistant to drugs that increase oxidative stress”. This sentence is not clear.  Cancer cells are more sensitive to antioxidant activity and do not have antioxidant capacity.

-          In the introduction section, the authors reported that Polyphenols, derived from plants, are well-known for their antioxidant properties, but their pro-oxidative activity may play a more significant role in cancer treatment. However, they should briefly discuss: (1) the differential effects of polyphenols on mitochondria of normal versus cancer cells; (2) the fact that in some studies the protective or pro-oxidant effect of polyphenols against cancer cells has been shown to be concentration-dependent.

-          In Table 1, the authors could also discuss the effect of other polyphenols such as Ginstein, Gallic Acid, Tannic acid, gossypol that are known to have a prooxidant effect on cancer cells.

-          The captions of the complex Figures 2-7 reproduced with permission  do not provide enough information.  Consequently, it is difficult to interpret the figures without consulting the original paper from which they were taken.

-          The authors could also discuss other nano carriers for polyphenol delivery such as nanoemulsions, nanogels, AuPNs

Comments on the Quality of English Language

The quality of English language is good. The manuscript is written clearly enough and requires Minor editing of English language.

Author Response

Comments 1: In this review the authors discuss the encapsulation strategies of polyphenols in nanocarriers with the aim of improve the stability and bioavailability of polyphenols, thus enhancing their antitumor effect by disrupting the mitochondria of tumor cells.

Although there are already other reviews on the topic, this one is focused on the regulation of RE-DOX mitochondria. It is interesting and could be published in this journal but should be enhanced by major revisions.

Specific comments:

  1. In the introduction section authors reported “Cancer cells with higher intrinsic levels of reactive oxygen species have a more sensitive antioxidant capacity than normal cells, making them less resistant to drugs that increase oxidative stress”. This sentence is not clear.  Cancer cells are more sensitive to antioxidant activity and do not have antioxidant capacity.

    Response 1: We thank the reviewer for this suggestion. This sentence has been revised to ‘Cancer cells with elevated levels of ROS possess a greater antioxidant load and a delicate redox equilibrium compared to normal cells, rendering them more susceptible to drugs that induce oxidative stress’.

    Comments 2: In the introduction section, the authors reported that Polyphenols, derived from plants, are well-known for their antioxidant properties, but their pro-oxidative activity may play a more significant role in cancer treatment. However, they should briefly discuss: (1) the differential effects of polyphenols on mitochondria of normal versus cancer cells; (2) the fact that in some studies the protective or pro-oxidant effect of polyphenols against cancer cells has been shown to be concentration-dependent. (3) In Table 1, the authors could also discuss the effect of other polyphenols such as Ginstein, Gallic Acid, Tannic acid, gossypol that are known to have a prooxidant effect on cancer cells.

    Response 2: We thank the reviewer for this suggestion.

    (1) Interestingly, polyphenols can selectively decrease cancer cells' vitality com-pared to normal cells [18-20]. Studies indicate that polyphenols can swiftly prompt ROS formation in cancer cells while sparing normal cells[19,20]. This imbalance between ROS production and antioxidant defense efficiency is a key factor. Polyphenols induced differential expression of antioxidant en-zymes such as catalase and superoxide dismutase in cancer cells and normal cells, respectively [21]. In cancer cells, there is a depletion of antioxidant pools, generation of free radicals, and collapse of mitochondrial membrane potential, while normal cells exhibit high levels of catalase and insensitivity to hydrogen peroxide[22].

    This section is added to Introduction in page 2.

    (2) Additionally, some studies have demonstrated that the pro-oxidant effect of polyphenols on cancer cells is dependent on concentration[23-25]. Although high concentrations of polyphenols can lead to oxidative stress and harm, lower levels can protect against H2O2-induced damage in cancer cells. Therefore, maintaining optimal polyphenol levels in vivo is crucial for their anti-cancer properties.

    This section is added to Introduction in page 2.

    (3) Genistein, Gallic Acid, Tannic acid, gossypol have been discussed in the Table 1 in page 6.

    Comments 3:The captions of the complex Figures 2-7 reproduced with permission  do not provide enough information.  Consequently, it is difficult to interpret the figures without consulting the original paper from which they were taken.

    Response 3: We thank the reviewer for this suggestion. The figures have been simplified and redone specifically. Enough information has been provided to describe Figures 2-7.

    Comments 4: The authors could also discuss other nano carriers for polyphenol delivery such as nanoemulsions, nanogels

    Response 4: We thank the reviewer for this suggestion. Other nano carriers for polyphenol delivery such as nanoemulsions, and nanogels, have also been developed to satisfy different anti-cancer strategies. Nanoemulsion, a distinctive structure consisting of an oil phase, water phase, and surfactant or emulsi-fier, is regarded as a delivery system for improving the efficacy of hydrophobic anti-tumor drugs[127]. Nanoemulsion can enhance the efficacy and stability of drug dos-age, as well as exhibit slow release and targeting effects. For instance, quercetin-nanoemulsion demonstrates improved encapsulation efficiency, and enhanced anti-tumor effects compared with free quercetin[128]. Nanogel has emerged as another promising nano carrier for the targeted delivery of therapeutic agents in cancer treat-ment[129]. With its porous structure and large surface-to-volume ratio, nanogel can enhance drug permeability and retention at the tumor site, thereby improving thera-peutic outcomes. A curcumin-loaded nanogel has achieved sustained drug release, leading to enhanced suppression of cancer cells and tumor growth compared to free curcumin[130].

    This section is added to 2.7 Other polyphenolic nanomedicine in line 352-364, page 20.

Round 2

Reviewer 1 Report

Comments and Suggestions for Authors

I'm OK with the changes made. 

Author Response

Comments 1: I'm OK with the changes made.

Response 1: We thank the reviewer for your recognition of the changes we have made. 

Reviewer 2 Report

Comments and Suggestions for Authors

Include this paper in your manuscript.

IJMS | Free Full-Text | Recent Advances in Synergistic Effect of Nanoparticles and Its Biomedical Application (mdpi.com)

Comments on the Quality of English Language

Revise it carefully.

Author Response

Comments 1: Include this paper in your manuscript. IJMS| Free Full-Text | Recent Advances in Synergistic Effect of Nanoparticles and Its Biomedical Application (mdpi.com)

Response 1: We have cited this paper in our manuscript (Reference No.69).

Response to Comments on the Quality of English Language:

Revise it carefully.

Response: We thank the reviewer for this suggestion. The English Language of the article has been revised by a medical professional.

Reviewer 4 Report

Comments and Suggestions for Authors

The authors have satisfactorily addressed my concerns and improved the quality of the manuscript.

I recommend publication of this manuscript in Pharmaceutics 

Author Response

Comments 1:The authors have satisfactorily addressed my concerns and improved the quality of the manuscript.

Response 1: We thank the reviewer for your recognition of the changes we have made.
